# Study of Dispersions of Carbon Nanotubes Modified by the Method of Rapid Expansion of Supercritical Suspensions

**DOI:** 10.3390/molecules25184061

**Published:** 2020-09-05

**Authors:** Konstantin B. Ustinovich, Victor V. Ivanov, Yurij M. Tokunov, Alexander A. Loshkarev, Natalya I. Sapronova, Anton M. Vorobei, Olga O. Parenago, Michael G. Kiselev

**Affiliations:** 1Kurnakov Institute of General and Inorganic Chemistry of the Russian Academy of Sciences, 119071 Moscow, Russia; ustinovich@supercritical.ru (K.B.U.); vorobei@supercritical.ru (A.M.V.); oparenago@scf-tp.ru (O.O.P.); 2Moscow Institute of Physics and Technology, National Research University, 141701 Dolgoprudny, Russia; ivanov.vv@mipt.ru (V.V.I.); tokunov.ium@mipt.ru (Y.M.T.); loshkarev.aa@phystech.edu (A.A.L.); Sapronova@phystech.edu (N.I.S.); 3Chemical Department, Lomonosov Moscow State University, 119234 Moscow, Russia; 4G.A. Krestov Institute of Solution Chemistry of the Russian Academy of Sciences, 153045 Ivanovo, Russia

**Keywords:** RESS, supercritical fluids, carbon nanotubes, ultrasound

## Abstract

The effectiveness of carbon nanotubes (CNT) deagglomeration by rapid expansion of supercritical suspensions (RESS) in nitrogen and carbon dioxide fluids was studied in this work. Two different mechanisms of deagglomeration were proposed for these two fluids at various temperature and pressure conditions. Ultrasound attenuation spectroscopy was applied as an express method of determining median diameter and aspect ratio of CNTs. At least twofold reduction of the diameter was shown for CNT bundles processed by RESS technique. Aspect ratio of processed CNTs, calculated from acoustic attenuation spectra, increased to 340. These results were in a good agreement with atomic force microscopy data.

## 1. Introduction

Uniqueness of physical properties of carbon nanotubes (CNTs) basically define by their thermodynamics state, namely, the most CNT applications in microelectronics [1], polymer nanocomposites [2,3], displays [4], solar panels [5], membranes [6], and many others [7,8], require stable homogeneous CNT dispersions without agglomerates or bundles. Unfortunately, CNTs, like all nano-sized objects, tend to aggregate. CNTs are poorly dispersible in most solvents: They form large aggregates and the stability of nanotube suspensions is usually low. The high hydrophobicity of the nanotube surface is also an important factor increasing their aggregation in water suspensions.

The methods of preparation of stable homogeneous CNT dispersions most frequently involve surfactants and ultrasonic treatment [9]. They often require continuous powerful sonication. Despite the high effectiveness of CNT deagglomeration, this method has a significant disadvantage: Such treatment can make CNTs much shorter, which leads to undesirable changes in the properties of the final product containing CNTs [10,11,12]. Besides, it may also result in more structural defects in the CNTs [13,14].

An alternative and “gentler” technique for processing CNTs can be the RESS (Rapid Expansion of Supercritical Suspensions) method. It is a modification of the more widely known method of Rapid Expansion of Supercritical Solutions, used for micronizing substances soluble in a certain supercritical fluid [15,16,17]. The Rapid Expansion of Supercritical Suspensions (RESS) method consists in the following: A suspension of the starting powdered material in a supercritical fluid (SCF) is produced in a suspension chamber at the preset parameters of state. After storing the suspension for the required amount of time at elevated pressure, it is rapidly sprayed into the precipitation chamber at atmospheric pressure. After the spraying, the fluid rapidly expands behind the nozzle and undergoes a transition from supercritical to gas phase, becomes less dense, and loses its dissolving capacity. The microstructure of the dispersed material changes, mainly due to the rapid and non-uniform pressure drop. The main operating parameters influencing the effectiveness of nanomaterial deagglomeration are the type and composition of the supercritical fluid, its pressure and temperature in the dispersion chamber, and processing time. Depending on the application purpose, an SCF-suspension can be sprayed into a gas to obtain a dry powder (the “dry” method) or into a solvent medium to form a CNT dispersion (the “wet” method) [18]. In the latter case, it is possible to add surfactants to the liquid medium for achieving additional stabilization of CNTs in the suspension. The suspension in this case is usually sprayed through a tube, immersed into the liquid from the bottom of the precipitation vessel, rather than through a nozzle mounted on top of the precipitation vessel or through a ball valve between the dispersion chamber and the precipitation vessel.

In this work, we have studied the characteristics of aqueous dispersions of two CNT types: Starting CNTs and CNTs modified by the method of Rapid Expansion of Supercritical Suspensions, RESS [19,20].

## 2. Materials and Sample Preparation Procedures

In the experiments, we used two CNT types: Starting single-wall TUBALL CNTs manufactured by OCSiAl (Novosibirsk, Russia) (CNTs-T) and CNTs modified by the Rapid Expansion of Supercritical Suspensions (RESS) method (CNTs-M). Sodium dodecylbenzenesulfonate (SDBS) was purchased by Sigma Aldrich (Steinheim, Germany) (technical grade).

The scheme of the RESS installation is shown in Figure 1. In this work, the RESS processing was carried out by the dry method. The pressure in the 500 ml precipitation vessel was atmospheric. Nitrogen and carbon dioxide were used as the fluids. The following parameters of state were applied: Pressure from 50 to 150 bar, temperature from 40 to 80 ℃ for the nitrogen, pressure from 100 to 300 bar, temperature from 40 to 100 ℃ for the carbon dioxide. The volume of suspension chamber 3 was 25 mL. 

Two types of liquid dispersions were prepared based on two types of CNTs (starting CNTs and nitrogen-modified ones). The starting CNTs were diluted with deionized water with an addition of a sodium dodecylbenzenesulfonate (SDBS) dispersant for steric stabilization of the nano-objects during the dispersion preparation procedure. To remove large agglomerates of nano-objects from the dispersions, we subjected the latter to intensive ultrasonic dispersing with an acoustic power of 100 W for 60 min and to accelerated sedimentation in a 5000 g centrifugal field for 60 min in a high-speed Sigma 3-30K centrifuge (Sigma Laborzentrifugen GmbH, Osterode am Harz, Germany) and then extracted the supernatant in an amount of 4/5 of the dispersion volume [21]. 

The final dispersion sample contained a smaller amount of carbon nanotubes than the starting mixture of water, a surfactant, and CNTs, since part of the solid phase precipitated after the accelerated sedimentation procedure. To determine the final concentration of the samples, we measured the coefficient of electromagnetic radiation absorption by the suspension at a wavelength of 260 nm with a Jasco V-770 spectrophotometer. The measurement results showed that the concentration of CNTs-T and CNTs-M in the dispersions was approximately the same (the concentration of CNTs-T was 3% lower than that of CNTs-M) and amounted to about 1.4 g/L.

The obtained dispersions were stable enough to be used in the study.

The following methods were applied for characterizing CNT dispersion samples: Transmission electron microscopy (TEM) on a JEM-2100 instrument at an accelerating voltage of 200 kV, scanning electron microscopy (SEM) on a JSM-7001F apparatus at an accelerating voltage of 30 kV, atomic force microscopy (AFM) in a semi-contact mode on an Ntegra Prima instrument, and acoustic spectroscopy (AS) on a DT-500 apparatus in the frequency range of 1–100 MHz.The samples for the microscopy studies were prepared by diluting the starting dispersions by about 1000 times and depositing them onto carbon networks for the TEM and onto atomically smooth substrates made of mono-crystal silicon steel for the SEM and AFM measurements.

The CNTs were also characterized by the Raman spectroscopy method with a Horiba LabRAM HR Evolution confocal Raman microscope with He-Ne laser excitation (λ = 632.8 nm).

## 3. Experimental Data and Discussion

When using the atomic force microscopy method, we measured the nanotube diameter and length based on the images obtained for each of the CNT samples to select at least 300 nanoobjects. The obtained data arrays for all the suspension samples are satisfactorily approximated by the probability density functions of the normal-logarithmic distribution with an average diameter d_N_ and an average length L_N_. The diameter and length values are shown in Table 1.

The characteristic AFM images of the nanotubes of the two types before and after RESS are shown in Figure 2. Figure 2 shows that RESS-processing leads to certain changes in the texture of the CNTs: the distance between the nanotubes increases. The AFM data in Table 1 also show that after the RESS-processing, the nanotube bundle diameter decreases from 5.3 nm to 2.5 nm, the nanotube length changes from 2200 nm to 970 nm, and the aspect ratio becomes only slightly lower decreasing from 415 to 388.

The typical examples of SEM images of the two nanotube suspension types before and after RESS are given in Figure 3. The SEM images also indicate that the RESS-processing leads to a certain change in the CNT texture at the mesoscopic level. The distance between the nanotubes increases, they become more ordered, and the material becomes looser at the microstructural level. AFM and SEM data indicate that RESS processing of CNTs leads to significant reduction of CNTs agglomeration degree. The main reasons of such qualitative changes in agglomerates structure can be associated with penetration of fluid into the gaps between the nanotubes and following depressurization with proppant effect of the expansion.

The typical examples of TEM images of the nano-objects of the two types of suspensions before and after RESS processing are presented in Figure 4. The TEM images also show that after the RESS-processing, the distance between the nanotubes increases, and they become more ordered. The starting nanotubes are packed in agglomerates that contain from 4 to several dozens of such tubes each. The CNTs after the RESS-processing are packed in agglomerates each containing from 2 to 10–15 nanotubes.

The TEM images indicate that the samples contain a large amount of amorphous carbon because the nanotube images are not sharp, and there are a lot of vortices, roughnesses, and lumps on the walls of the tubes. The samples also contain spherical inclusions with a diameter of several nanometers. An elemental analysis of these inclusions shows that they are the residue of the iron catalyst used for the CNT synthesis.

The biggest macroscopic change in the CNT samples after the RESS-processing is a considerable increase in the specific volume that represents a ratio of the CNT volume to its weight (Figure 5). The degree of the CNT specific volume increase after the RESS is equal to the ratio of the CNT specific volume after the RESS to its volume before the RESS treatment.

Depending on the RESS conditions, the increase in the specific volume of CNTs can vary within a wide range of values. The obtained ratios of the increased specific volume of the CNTs to its initial value depending on the RESS conditions are given in Figure 6.

When CO_2_ is used as the fluid at a pressure of 200 bar and the temperature rises from 40 to 100 °C, the CNT specific volume increase ratio goes up from 1.7 to 9.4. When N_2_ is used as the fluid at a temperature of 40 °C and the pressure goes up from 50 to 150 bar, the CNT specific volume increase ratio rises from 6.1 to 11.

It can be said that the parameters of state of the two fluids, N_2_ and CO_2_, have different effects on the volume increase ratio. In case of nitrogen, at higher pressure values, SCF-suspension processing leads to an increase in the specific volume regardless of the temperature over the whole range of values. At higher temperature values, the growth is more gradual than at low ones. The temperature in the range of 40–80 °C does not significantly increase the specific volume or have a great effect on the volume increase ratio. It can probably be assumed that in case of nitrogen, the main factor determining the degree of fluffing is atomization hydrodynamics. The higher the SCF-suspension treatment pressure, the greater is the pressure drop in case of spraying into a precipitation vessel under atmospheric pressure and, consequently, the bigger is the proppant effect of the expansion. When the fluid penetrates the CNT bundles, they are burst by the released expanding gas flows: The more the pressure drop in the chambers, the bigger is the distance between the bundle agglomerates. This is also confirmed by a direct correlation between the change in the pressure and the volume increase ratio and the fact that this effect is practically independent of the pressure. 

In the case of CO_2_, the dependences are more complex. The isothermal pressure increase in this fluid, in contrast to nitrogen, leads to a decrease in the specific volume increase ratio, and the isobaric temperature growth—to its increase. Obviously, in case of carbon dioxide, the process is driven by other factors. We assume that one of these factors is the phase state of CO_2_ during the expansion. The suspension processing conditions (100–300 bar, 40–100 °C) are quite close to the critical parameters of CO_2_ (74 bar, 31 °C), in contrast to those for nitrogen (34 bar, −147 °C). A rapid expansion is very likely to result in CO_2_ transition from this interval of parameters of state into a two-phase liquid–vapor system rather than to the gas phase. Such system must have a negative effect on the process of CNT bundle deagglomeration, since the appearing capillary effects lead to a pore collapse and nanotube agglomeration. The denser is the sprayed fluid, the higher is the liquid phase proportion. This hypothesis agrees well with the observed trends: the fluid density increases at the isothermal pressure growth and falls at the isobaric temperature increase. This hypothesis explains the observed results. The authors of work [22] came to similar conclusions concerning multi-walled CNTs. 

The CNT Raman spectra before and after RESS are shown in Figure 7. The two bands observed at 125 and 150 cm^−1^ correspond to the CNT radial breathing modes. All the samples that underwent supercritical treatment have a slightly higher ratio of the signal amplitudes, A(ν = 150 cm^−1^)/A(ν = 125 cm^−1^), than the starting CNTs, but no radical changes in this characteristic are observed. Moreover, it is possible to determine the diameter of individual tubes based on the frequency values of the breathing mode band by the empirical formula presented in work [23]: $D=\frac{\propto }{\nu_{\mathrm{BM}}}$, where α = (232 ± 10) cm^−1^ nm is the empirical factor, $\;\nu_{\mathrm{BM}}$ is the breathing mode frequency. The nanotube diameter determined from this ratio is equal to (1.55 ± 0.6) nm, which agrees with the TEM results and data provided by the OCSiAl company, the nanotube producer. The pronounced band at 1560 cm^−1^ corresponds to the so-called tangential modes. The distortion of these modes indicates defects in the nanotube structure. As Figure 7 shows, none of the tangential mode bands of the considered nanotubes indicate significant structural defects.

After the RESS-processing, we applied another method of analysis—acoustic spectroscopy—to obtain additional data about the bundle diameter and nanotube aspect ratio. This method has been widely used in the last few years to study opaque liquid dispersions with non-spherical particles [21]. This method registers ultrasound attenuation spectra in a dispersion medium in two dispersion states: In the state of rest with the isotropic angular distribution of an array of non-spherical nano-objects and in the stationary dispersion flow with the rotation axis of the array of cylindrical nano-objects parallel to the flow due to the flow acceleration in the contracting channel. The ultrasound attenuation spectra were measured in the frequency range of 3–100 MHz for two states of the dispersion: At the isotropic angular distribution of the nanotubes,$\;\alpha_{o}\left(\nu \right)$, and at the nanotube alignment in the direction perpendicular to the ultrasound wave $\alpha_{⊥}\left(\nu \right)$. Figure 8 shows the pairs of attenuation spectra $\alpha_{o}\left(\nu \right)$ and $\alpha_{⊥}\left(\nu \right)$ for the two types of the dispersions studied. The spectra reflect only the nanotube (nanoobject) contribution to the attenuation, with the additive aqueous medium contribution excluded from them. When plotting each of the spectral curves, we averaged at least three measurement results with good repeatability in order to reduce the measurement error.

The ultrasound attenuation spectra were used to determine the diameters of the CNT bundles; the aspect ratio for the two types of dispersions was obtained by the methods described in [21]. To estimate the diameter of nanotubes, the attenuation spectra of ultrasound on nanotubes predominantly aligned along the flow, perpendicular to the ultrasonic wave direction were used. The diameter of nanotubes was calculated by the DT-500 software algorithm described in [24]. According to this method the interaction of ultrasound and nanotubes interpreted as interaction with a model of nanotube—a chain a certain number of equivalent spheres arranged in cylinders. The validity of such a model proved on the one hand by the invariance of the volume of nano-objects, and on the other hand the invariance of the volume-specific surface [21].

The first condition is equivalent to the equality of the volume of a cylinder with diameter *d_AS_* and length *L* to the sum of the volumes of *N* spheres with diameter a:(1)$$2a^{3}N=3Ld_{AS}^{2}.$$

The second condition is equivalent to the equality of the areas of the surfaces of a cylinder and a set of *N* equivalent spheres and is expressed by the equation
(2)$$a^{2}N\approx Ld_{AS}.$$

Equations (1) and (2) give a simple relationship between the nanotube diameter and the diameter of equivalent spheres:(3)$$3d_{AS}=2a.$$

The aspect ratio of nanotubes was determined from the relative difference in the attenuation spectra of ultrasound for the isotropic and oriented states [21]. Averaging this difference over the measured frequency range ν_max-ν_min is represented by the integral:(4)$$\Phi \left(A\right)=\frac{1}{\nu_{max}-\nu_{min}}\int_{\nu_{min}}^{\nu_{max}}\frac{\alpha_{o}\left(\nu \right)-\alpha_{⊥}\left(\nu \right)}{\alpha_{⊥}\left(\nu \right)}d\nu .$$

The aspect ratio was calculated using the calibration dependence of the integral *Φ* (A) on the aspect ratio from [21]. The attenuation coefficient obtained at a random alignment of the nanotubes (the red lines in Figure 6) in the dispersion with modified CNTs is approximately 23% lower than in the dispersion with the starting CNTs. This is an indicator of the easier dispersibility of the CNT material in a liquid after RESS processing than with the starting CNTs.

The CNT-M aspect ratio (after RESS processing) calculated by the technique described in [21] based on the data of the ultrasound attenuation spectra in a CNT dispersion equaled 340, which agrees with the aspect ratio of 388 obtained by the atomic force microscopy method. The diameter of the CNT-M bundles after RESS processing was about 7.2 nm and was approximately 50% smaller than the diameter of the CNT-T bundles. This ratio of the bundle diameters agrees with the AFM data, which show that the value of this ratio is 2.1.

## 4. Conclusions

The conclusion of this work is that the RESS process is effective method of CNTs dispersing. Modification of the starting carbon nanotubes by the supercritical fluid technology method (the RESS method) makes it possible to obtain a stable dispersion of carbon nanotubes with a nanotube bundle diameter of about 7.2 nm in the dispersion, which may be significant for nanotube dispersion applications. Such conclusion is confirmed with different physico-chemical methods.The SEM images show that the network of modified CNTs has a noticeably larger cell size than that of the starting CNTs.A comparison of the results of the AFM measurements of the carbon nanotube diameters shows that the diameter of the modified carbon nanotube bundles is less than half of that of the starting carbon nanotube bundles.A comparison of the results of the carbon nanotube bundle diameter measurements in a dispersion by the method of acoustic spectroscopy shows that the diameter of the modified carbon nanotube bundles is less than half of that of the starting carbon nanotube bundles.A comparison of the ultrasound attenuation spectra in the two types of CNT dispersions shows that the attenuation coefficient in the dispersion with modified carbon nanotubes is approximately 23% lower than in the dispersion with the starting carbon nanotubes.

## Figures and Tables

**Figure 1 molecules-25-04061-f001:**
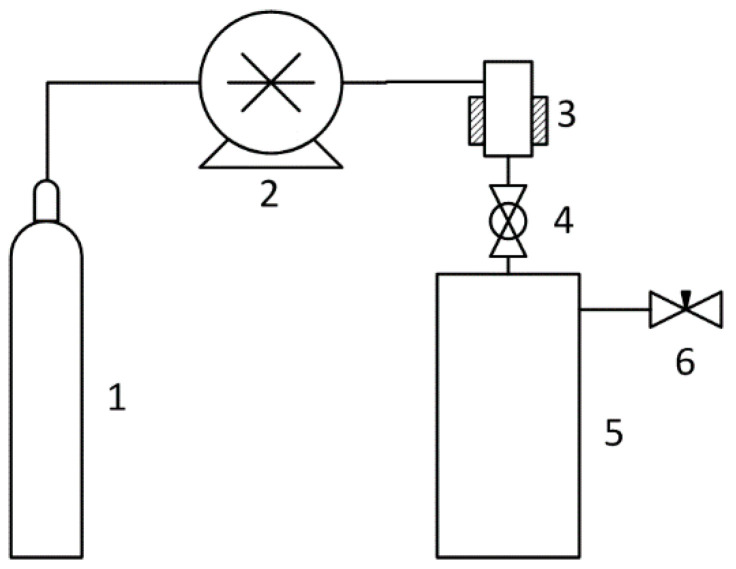
The scheme of the apparatus for rapid expansion of supercritical suspensions: 1 is the CO_2_ source, 2 is the CO_2_ pump, 3 is the suspension chamber, 4 is the 1.5 mm spray nozzle, 5 is the atmospheric pressure precipitation vessel, and 6 is the valve.

**Figure 2 molecules-25-04061-f002:**
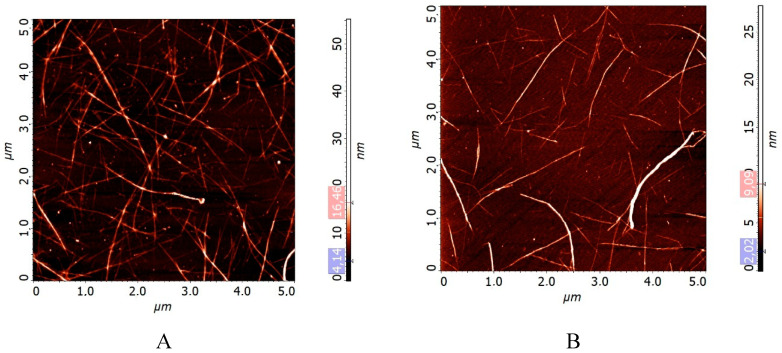
Carbon nanotubes atomic force microscopy (CNT AFM). Before (**A**) and after (**B**) CO_2_ processing at a pressure of 75 bar and a temperature of 80 °C.

**Figure 3 molecules-25-04061-f003:**
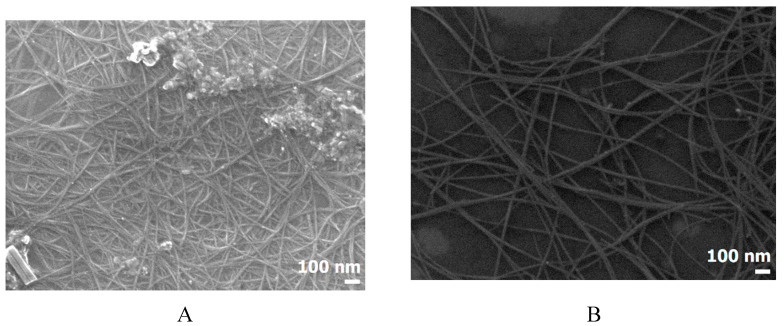
CNT SEM images. before (**A**) and after (**B**) CO_2_ processing at a pressure of 75 bar and a temperature of 80 °C.

**Figure 4 molecules-25-04061-f004:**
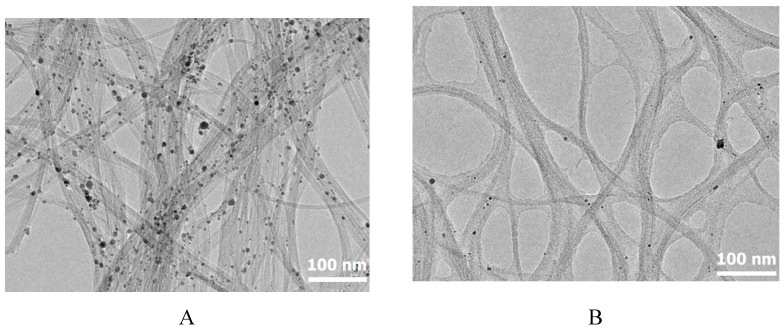
CNT TEM images. before (**A**) and after (**B**) CO_2_ processing at a pressure of 75 bar and a temperature of 80 °C.

**Figure 5 molecules-25-04061-f005:**
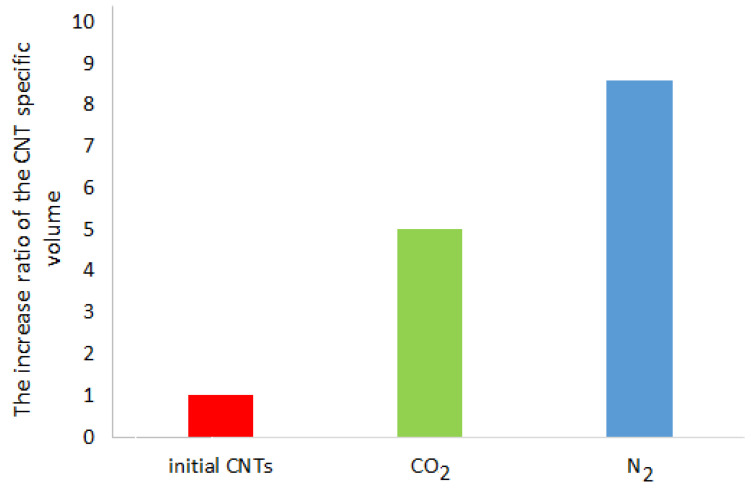
The increase ratio of the CNT specific volume after RESS (Rapid Expansion of Supercritical Suspensions)-processing (100 bar, 40 °C) using CO_2_ and N_2._

**Figure 6 molecules-25-04061-f006:**
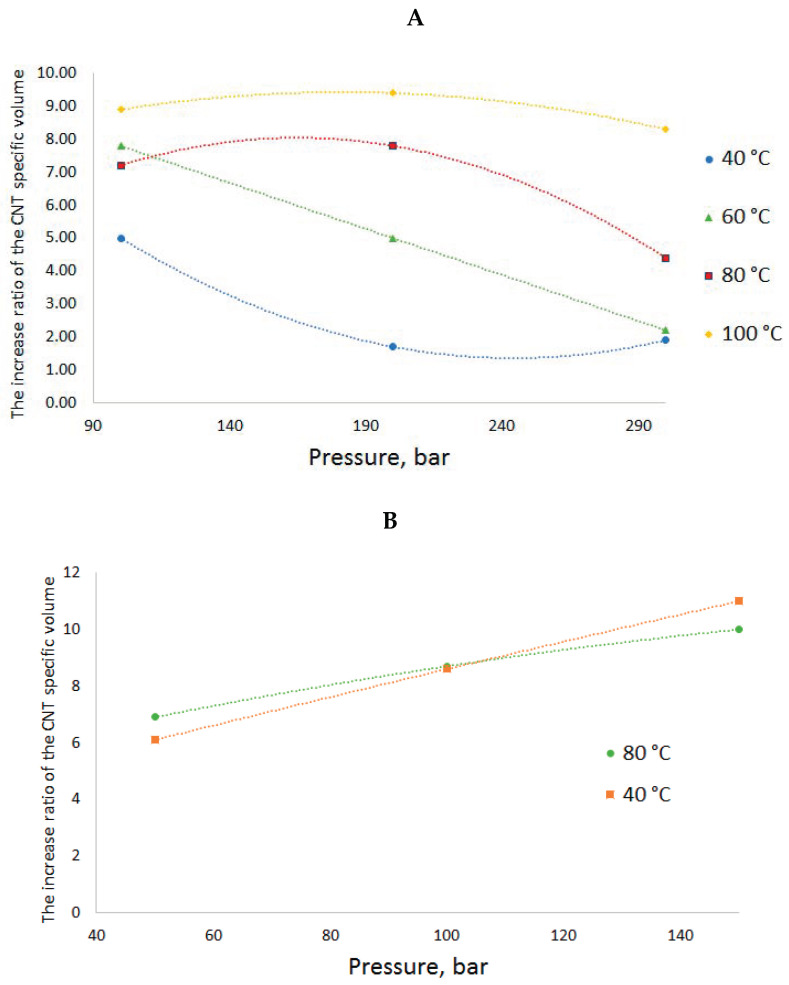
The increase ratio of the CNT specific volume after RESS-processing using CO_2_ (**A**) and N_2_ (**B**) at different conditions.

**Figure 7 molecules-25-04061-f007:**
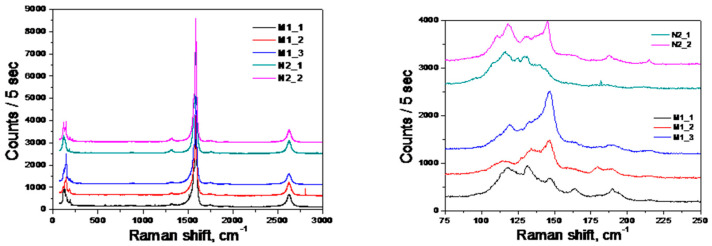
CNT Raman spectra before and after RESS-processing. N2_1 and N2_2 refers to the starting nanotubes, M1_1, M1_2, and M1_3—to the nanotubes after the RESS-processing with CO_2_ at a pressure of 75 bar and temperature of 80 °C.

**Figure 8 molecules-25-04061-f008:**
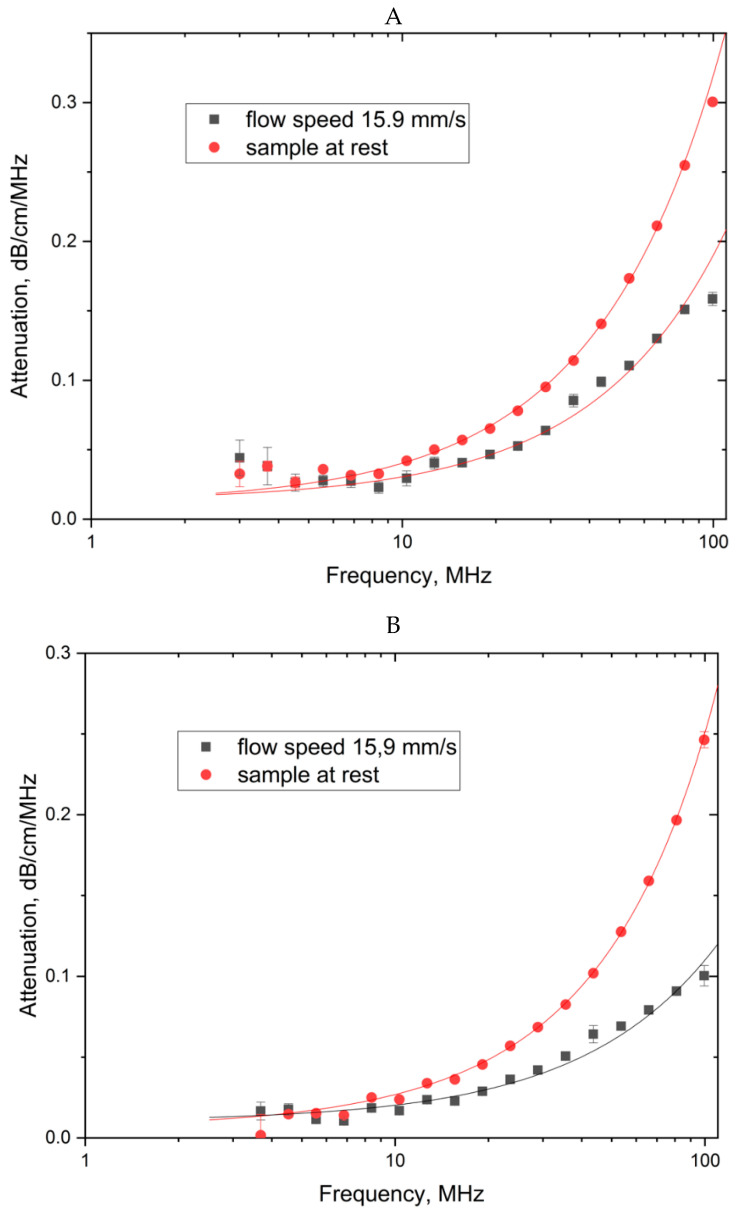
Ultrasound attenuation spectra measured in a CNT dispersion: The red lines denote the spectra obtained at a random alignment of the nanotubes, the black ones show the spectra reflecting the nanotubes predominantly aligned along the flow, perpendicular to the ultrasonic wave direction. (**A**)—CNTs before RESS, (**B)**—CNTs after RESS.

**Table 1 molecules-25-04061-t001:** Diameter, length and aspect ratio (A) of carbon nanotube bundles (CNTs-T and CNTs-M) in dispersions.

	AFM (Atomic Force Microscopy) Data		
	d_N_, nm	L_N_, nm	A
CNTs-T	5.3	2200	415
CNTs-M	2.5	970	388

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
