# Peer review of "Study of Dispersions of Carbon Nanotubes Modified by the Method of Rapid Expansion of Supercritical Suspensions"

_molecules, 2020, doi:10.3390/molecules25184061_

Round 1

Reviewer 1 Report

1) The 1st sentence should be re-writen - it is lazy and is essentially the same as every other CNT paper. Start with the statememnt of why what you are doing is important - which it is

2) Ref 14 it should be H2O2 not H2O2.

3) Figure and Legend should be on same pages - remember this is camera ready - also Page 3 lines 107 - 108 - dont use 1 sentence paragraphs

4) Multiple places CO2 not CO2.

5) cm-1 not cm-1.

6) Fig 6 would be better if before and after at a flow was in 1 chart and before and after at rest was in another - so the before and after could be compared on same scale

7) Conclusion section is a summary NOT a Conclusion - need to rewrite so it is a conclusion "The conclusion of this work is that the process is (good/bad) at dispersing....."

Author Response

Review Report 1

The authors thank the reviewer for his work and valuable comments. The authors agree with reviewer's remarks. Appropriate changes have been made:

  • «The 1st sentence should be re-writen - it is lazy and is essentially the same as every other CNT paper. Start with the statememnt of why what you are doing is important - which it is»

The first sentence is rewritten in accordance with reviewer’s comments.

2) «Ref 14 it should be H2O2 not H2O2.»

H2O2 was replaced with H2O2 (Ref. 14).

3) «Figure and Legend should be on same pages - remember this is camera ready - also Page 3 lines 107 - 108 - dont use 1 sentence paragraphs»

Appropriate changes have been made.

4) «Multiple places CO2 not CO2.»

CO2 was replaced with. CO2.

5) «cm-1 not cm-1.»

cm-1 was replaced with cm-1.

6) «Fig 6 would be better if before and after at a flow was in 1 chart and before and after at rest was in another - so the before and after could be compared on same scale»

Such configuration of spectra would be not informative, because it is important to compare the ultrasound attenuation spectra in the case of two states of the dispersion: at the isotropic angular distribution of the nanotubes and at the nanotube alignment in the direction perpendicular to the ultrasound wave. Only using this pair of spectra it is possible to determine the diameters of the CNT bundles. [V.V.Ivanov, A.A.Loshkarev, M.F.Vlasova, A.A. Lizunova, N.I.Sapronova, Yu.M.Tokunov. Acoustic spectroscopy for evaluating dimensions of cylindrical carbon nano-objects in colloidal systems//Colloids and Surfaces A: Physicochemical and Engineering Aspects. 2017. v.520. p. 640-648.]

7) «Conclusion section is a summary NOT a Conclusion - need to rewrite so it is a conclusion "The conclusion of this work is that the process is (good/bad) at dispersing.....»

The text was changed according to reviewer's remark.

Reviewer 2 Report

In this study, the authors suspended carbon nanotubes (CNT) in water by using a rapid expansion process of supercritical suspensions (RESS). Two different supercritical fluids, nitrogen and carbon dioxide, was studied in this work to understand the effectiveness of suspension stability at various temperature and pressure conditions. Ultrasound attenuation spectroscopy was applied to probe diameter and aspect ratio of suspended CNTs. The results showed that CNT bundles processed by RESS technique can result in a larger aspect ratio of 340. Overall, the results were new and the findings provided certain scientific insights, but a lot of experimental details were missing. More details should be provided with in-depth discussions. My suggestions and comments are listed below for the authors to perfect the article:

  1. The source of CNTs and surfactants should be provided Were these used as purchased or purification were needed?
  2. The size of the suspension chamber, nozzle, or other devices in Figure 1 should be provided or briefly descried.
  3. The pictures of CNT/water solutions before and after process should be provided to convince reader how the solutions were stabilized.
  4. In Table 1, the authors should address the abbreviation of ASM.
  5. Scale bar in Figure 2&4 is not clear. Please revise. It is too hard for readers to see the differences. Please either enlarge the images or provide a better resolution.
  6. The mechanism of the bundle de-agglomeration should be addressed with more details in the discussion of figure 2&3.
  7. Can the authors present the data in Table 2 as figures? It would be easier to compare the differences at various temperatures and pressures.
  8. The legends in Figure 5 are messy. Although the caption indicates M1, but what does M1_1, M1_2, and M1_3 means? Moreover, why are there two N2_1?Please revise.
  9. How can the authors extract information from the measurements in Figure 6? The calculation or fitting equations should be addressed in detail.

Author Response

Review Report 2

The authors thank the reviewer for his work and valuable comments. The authors agree with reviewer's remarks. Appropriate changes have been made:

  1. «The source of CNTs and surfactants should be provided Were these used as purchased or purification were needed?»

The source of CNTs and surfactants was provided. These substances were used as purchased without additional purification.

  1. «The size of the suspension chamber, nozzle, or other devices in Figure 1 should be provided or briefly descried.»

The size of the suspension chamber, nozzle etc in Figure 1 were provided.

  1. «The pictures of CNT/water solutions before and after process should be provided to convince reader how the solutions were stabilized.»

The authors did not provide such pictures, because they are not informative. CNT/water solutions are black suspensions and it is difficult to demonstrate any difference before and after process visually.

  1. «In Table 1, the authors should address the abbreviation of ASM.»

ASM was replaced with AFM (atomic force microscopy)  In Table 1

  1. «Scale bar in Figure 2&4 is not clear. Please revise. It is too hard for readers to see the differences. Please either enlarge the images or provide a better resolution.»

Scale bar in Figure 2&4 were modified according to reviewer's remark.

  1. «The mechanism of the bundle de-agglomeration should be addressed with more details in the discussion of figure 2&3.»

Discussion of figure 2&3 was added to the text.

  1. «Can the authors present the data in Table 2 as figures? It would be easier to compare the differences at various temperatures and pressures.»

Data of Table 2 were presented as figures.

  1. «The legends in Figure 5 are messy. Although the caption indicates M1, but what does M1_1, M1_2, and M1_3 means? Moreover, why are there two N2_1?Please revise.»

The legends in Figure 5 were modified according to reviewer's remark.

  1. «How can the authors extract information from the measurements in Figure 6? The calculation or fitting equations should be addressed in detail.»

Additional discussion was added to the text.

Round 2

Reviewer 2 Report

The authors have revised the manuscript properly according to the review comments. It is now in a good shape for publication.